# Differences in natriuretic peptide response in self-identified white and black individuals: a physiological clinical trial

Black individuals have lower plasma natriuretic peptide (NP) concentrations than white individuals. However, race-based differences in the NP response to physiological perturbations are unknown. In this physiological trial (NCT#03070184), we measured the NP [mid-regional atrial NP (MR-proANP), N-terminal pro-B-type NP (NT-proBNP), and BNP] response to physiological perturbations among healthy, self-identified Black and white participants aged 18-40 years. The primary and secondary outcomes were the change in plasma NP concentrations at 6 weeks after metoprolol (initiated at 50 mg/day and doubled every 2 weeks) and standardized, aerobic exercise (70% of their maximal oxygen uptake on a salt-standardized background), respectively. Among 40 Black [median age: 27 (22, 32) years; 21 (52.5%) women] and 40 white [median age: 25 (20, 30) years; 19 (47.5%) women] participants, exercise increased MR-proANP (Black: 35%; white: 43%), NT-proBNP (Black: 11%; white: 23%), and BNP (Black: 59%; white: 61%) in both self-reported races. Exercise was associated with an increase in plasma MR-proANP ($p_{interaction}$: 0.25) and BNP ($p_{interaction}$: 0.87) concentrations which did not vary by self-reported race. However, the increase in plasma NT-proBNP concentrations were higher in white participants than in Black participants. ($p_{interaction}$: 0.04) Similarly, metoprolol therapy increased MR-proANP (Black: 18%; white: 16%), NT-proBNP (Black: 95%; white: 99%), and BNP (Black: 45%; white: 74%) in both self-reported races. The metoprolol-associated increase in plasma MR-proANP ($p_{interaction}$: 0.85), NT-proBNP ($p_{interaction}$: 0.94), and BNP ($p_{interaction}$: 0.21) concentrations were similar by self-reported race. In conclusion, the higher increase in plasma NT-proBNP concentrations among white patients after exercise suggests that exercise may induce significant physiological variations in NP levels. ClinicalTrials.gov ID: NCT03070184.

Natriuretic peptides (NPs) are a group of cardiometabolic hormones secreted by the heart[1,2]. In healthy individuals, NPs regulate a range of physiological functions such as vasodilation, natriuresis, glucose utilization, and lipid metabolism[3,4]. The concentrations of these bioactive peptides are influenced by factors such as body mass index, sex, and race[5–7]. NP concentrations have been shown to be lower in Black individuals than their white counterparts[5,8–12]. Population-level data has supported that these racial differences in NP concentrations begin in

✉e-mail: parora@uabmc.edu

childhood and persist across the age range[11]. Furthermore, circulating NP concentrations decrease with increasing proportion of African ancestry among Black individuals[8,9].

Low concentrations of NPs have also been associated with cardiometabolic diseases such as hypertension, diabetes, and obesity[3,4,13,14]. The role of low NP concentrations in the development of cardiometabolic diseases has been delineated through multiple lines of evidence, including epidemiological, animal, and genetic studies[13–18]. However, the NP system has been shown to be responsive to perturbations. Physiological stimuli such as exercise[19–21] and pharmacological interventions such as beta-blockers[22–25] have been shown to increase NP concentrations. The prior studies were limited by their assessment of NP response to metoprolol in patients with cardiovascular disease[25–27]. Furthermore, the examination of the NP response to the abovementioned perturbations did not take racial differences in the NP system into consideration. Black individuals have lower NP concentrations, which have been attributed to differences in the processing and degradation of NPs by race[7–11]. Additionally, the hemodynamic response to exercise and recovery post-exercise also vary by race[28]. Hence, it could be postulated that the responsiveness of the NP system to perturbations may also vary by race. However, data on the NP response to perturbation in Black individuals and their comparison with white individuals is lacking.

Therefore, this study hypothesized that the NP system in Black individuals will be less responsive to perturbations, including exercise and beta-blockers, compared with white individuals. The current prospective clinical trial accounted for factors affecting NP concentrations by recruiting healthy young participants without obesity with a similar representation of women in both groups. This prospective, single-center clinical trial aimed to compare the NP response to a standardized exercise challenge and 6 weeks of metoprolol. Here, we show that plasma NP levels increase in young, healthy white and Black participants in response to standardized exercise and metoprolol therapy. However, the increase in the N-terminal-pro-B-type NP (NT-proBNP) concentrations with exercise was greater in white individuals compared with their Black counterparts, highlighting that physiological perturbations may have a higher level of NP augmentation in white individuals compared with Black individuals.

## Results

A total of 112 participants were screened for the study from 2018 to 2023 at the University of Alabama at Birmingham. Of the 112 participants, 20 were ineligible for the study, 87 completed the exercise phase of the study, and 83 completed the medication phase of the study. There were 10 participants who completed only a single study phase. The final study sample consisted of 80 individuals, 40 were self-reported white individuals and 40 were self-reported Black individuals (Supplementary Fig. 1).

Baseline characteristics stratified by self-reported race are described in Table 1. Both groups had a similar percentage of women (52.5% in Black individuals and 47.5% in white individuals), BMI [25.1 (22.4, 27.7) kg/m$^2$ in Black individuals and 24.1 (22.4, 26.8) kg/m$^2$ in white individuals], median age [27 (22, 32) years in Black individuals and 25 (20, 30) years in white individuals), and mean systolic BP (110.7 ± 9.9 mm Hg in Black individuals and 109.4 ± 9.3 mm Hg in white individuals). (Table 1) At baseline, 13 participants had plasma NT-proBNP concentrations below the detectable limit. Of these 13 participants, 8 were Black individuals and 5 were white individuals.

### Differences in NP response to a standardized exercise challenge based on self-reported race

In the overall population, plasma mid-regional pro-atrial natriuretic peptide (MR-proANP) concentrations increased by 39% (41 ng/L at baseline to 57 ng/L after exercise), plasma BNP concentrations increased by 60% (15 ng/L at baseline to 25 ng/L after exercise), and

plasma NT-proBNP concentrations increased by 17% (19 ng/L at baseline to 22 ng/L after exercise) after exercise.

On stratification by self-reported race, the increase in MR-proANP was 43% (43 ng/L at baseline to 61 ng/L immediately after exercise), BNP was 61% (15 ng/L at baseline to 25 ng/L immediately after exercise), and NT-proBNP was 23% (23 ng/L at baseline to 28 ng/L immediately after exercise) immediately after exercise among white individuals (Supplementary Fig. 2).

Among Black individuals, the increase in MR-proANP, BNP, and NT-proBNP was 35% (39 ng/L at baseline to 52 ng/L immediately after exercise), 59% (16 ng/L at baseline to 25 ng/L immediately after exercise), and 11% (16 ng/L at baseline to 18 ng/L immediately after exercise), respectively (Supplementary Fig. 2).

The increase in MR-proANP ($p_{interaction}$:0.25) and BNP ($p_{interaction}$:0.87) did not vary by self-reported race. However, the increase in NT-proBNP was higher in white individuals than in Black individuals ($p_{interaction}$:0.04).

Figure 1 depicts the change in absolute NP values with standardized exercise stratified by self-reported race.

The urinary sodium excretion pre-exercise and post-exercise were similar in Black ($n = 28$) [109.4 (9.2) mmol/L pre-exercise to 102.4 (7.5) mmol/L post-exercise; p:0.21] but the urinary sodium excretion decreased post-exercise in white ($n = 35$) [108.6 (11.1) mmol/L pre-exercise to 83.7 (8.8) mmol/L post-exercise; p:0.02] individuals (Supplementary Fig. 3).

The heart rate and blood pressure before, immediately, 15 minutes, and 30 minutes after exercise stratified by self-reported race have been depicted in Supplementary Fig. 4.

### Differences in NP response to metoprolol based on self-reported race

An increase in MR-proANP by 17% (41 ng/L at baseline to 48 ng/L at 6 weeks; $p_{time}$: < 0.001), BNP by 59% (15 ng/L at baseline to 25 ng/L at 6 weeks; $p_{time}$: < 0.001), and NT-proBNP by 97% (19 ng/L at baseline to 37 ng/L at 6 weeks; $p_{time}$: < 0.001) was noted after 6 weeks of metoprolol in the overall population.

Among white individuals, treatment with 6 weeks of metoprolol leads to a 16% increase in plasma MR-proANP concentrations (43 ng/L at baseline to 51 ng/L at 6 weeks; $p_{time}$: < 0.001), 74% increase in plasma BNP concentrations (15 ng/L at baseline to 27 ng/L at 6 weeks; $p_{time}$: < 0.001), and 99% increase in plasma NT-proBNP concentrations (22 ng/L at baseline to 45 ng/L at 6 weeks; $p_{time}$: < 0.001) (Supplementary Fig. 5).

Plasma MR-proANP concentrations increased 18% (39 ng/L at baseline to 46 ng/L at 6 weeks; $p_{time}$: < 0.001), plasma BNP concentrations increased 45% (16 ng/L at baseline to 23 ng/L at 6 weeks; $p_{time}$: < 0.001), and plasma NT-proBNP concentrations increased by 95% (16 ng/L at baseline to 31 ng/L at 6 weeks; $p_{time}$: < 0.001) after 6 weeks of metoprolol in Black individuals (Supplementary Fig. 5).

However, the increase in MR-proANP ($p_{interaction}$:0.85), BNP ($p_{interaction}$:0.21), and NT-proBNP ($p_{interaction}$:0.94) did not vary by self-reported race.

Figure 2 depicts the change in absolute NP values with 6 weeks of metoprolol stratified by self-reported race.

### Change in blood pressure and heart rate with metoprolol

In the overall population, the systolic BP [108 (9.6) mm Hg at baseline to 102 (9.0) mm Hg at 6 weeks] and heart rate [67 (11) bpm at baseline to 57 (9) bpm at 6 weeks] decreased after 6 weeks of metoprolol therapy (Fig. 3). The decrease in systolic BP at 6 weeks was larger in white individuals [107 (10) mm Hg at baseline to 99 (8) mm Hg at 6 weeks] than in Black individuals [109 (9) mm Hg at baseline to 105 (9) mm Hg at 6 weeks] (Fig. 3). The decrease in heart rate at 6 weeks was similar in white individuals [65 (10) bpm at baseline to 55 (8) bpm at 6 weeks] than in Black individuals [69 (12) bpm at baseline to 59

(9) bpm at 6 weeks] (Fig. 3). However, the decrease in systolic BP (p$_{interaction}$: 0.13) and heart rate (p$_{interaction}$: 0.68) did not vary by self-reported race.

### Change in 24-hour urinary sodium excretion and serum insulin with metoprolol

Total 24-hour urinary sodium excretion did not decrease with 6 weeks of metoprolol therapy in black ($n = 29$) [129.0 (10.5) mmol/day at baseline to 130.3 (13.1) mmol/day at 6 weeks; p:0.93] and white ($n = 35$) [137.1 (10.7) mmol/day at baseline to 119.2 (11.5) mmol/day at 6 weeks; p:0.22] individuals (Supplementary Fig. 3). Even though Black participants had higher serum insulin concentrations at baseline than white participants, the change in serum insulin concentrations with metoprolol was similar in Black and white participants (Supplementary Fig. 6).

The differences in the change in HOMA-IR and serum aldosterone levels with metoprolol have been depicted in Supplementary Figs. 7–8.

The results of the sensitivity analysis have been presented in the Supplementary Results.

## Discussion

This physiologically rigorous clinical trial incorporating salt-standardized diets in young, healthy, normotensive adults assessed the NP response to acute aerobic exercise and metoprolol. Exercise led to an increase in the plasma MR-proANP, NT-proBNP, and BNP concentrations, with the most notable increase in plasma BNP concentrations. Standardized exercise was associated with a significantly higher increase in plasma NT-proBNP concentrations in white participants (23%) than in Black participants (11%). Although not statistically significant, the increase in plasma MR-proANP concentrations immediately after exercise was also higher in white participants (43%) than in Black participants (35%). This study showed that the increase in plasma BNP concentrations with standardized exercise was similar in Black and white individuals. Similarly, plasma MR-proANP, NT-proBNP, and BNP concentrations increased significantly with 6 weeks of metoprolol therapy. The 6-week course of metoprolol was well-tolerated by the participants. Although the differences were not significant, increasing doses of metoprolol led to a larger decrease in heart rate and systolic BP in white individuals than in Black individuals. The increase in all 3 NP subtypes was similar in both groups. NP augmentation with exercise and metoprolol was associated with a decrease in serum insulin concentrations, which was similar in Black and white participants. To summarize, even though Black individuals are known to have lower plasma NP concentrations, the increase in plasma NP concentrations in Black young adults to metoprolol was similar to their white counterparts. However, the NT-proBNP and MR-proANP response to exercise was found to be higher in white participants compared with Black participants.

### Table 1 | Baseline characteristics stratified by race

|  | Black participants ($n = 40$) | White participants ($n = 40$) | *p*-value |
|---|---|---|---|
| **Age, years** | 27 (22, 32) | 25 (20, 30) | 0.10 |
| **Women, *n* (%)** | 21 (52.5%) | 19 (47.5%) | 0.66 |
| **Body Mass Index, kg/m²** | 25.1 (22.4, 27.7) | 24.1 (22.4, 26.8) | 0.58 |
| **SBP, mmHg** | 109 (105, 116) | 108 (102, 116) | 0.55 |
| **DBP, mmHg** | 67 (64, 74) | 70 (63, 75) | 0.48 |
| **Heart rate, beats per minute** | 69 (63, 77) | 70 (62.7, 79) | 0.74 |
| **Fasting Glucose, mg/dL** | 83.0 (80.0, 89.0) | 86.5 (79.0, 90.0) | 0.54 |
| **Hemoglobin, g/dL** | 13.3 (12.5, 14.1) | 14.1 (12.6, 14.8) | 0.08 |
| **Insulin, mg/dL** | 6.5 (3.9, 9.0) | 5.9 (3.7, 7.9) | 0.25 |
| **Creatinine, mg/dL** | 1.0 (0.8, 1.1) | 0.9 (0.7, 1.0) | 0.11 |
| **AST U/L** | 17.0 (15.0, 25.0) | 17.0 (14.5, 21.0) | 0.61 |
| **ALT U/L** | 14.0 (11.0, 18.0) | 14.5 (12.0, 20.5) | 0.76 |
| **MR-proANP, pmol/L** | 43 (31, 56) | 47 (35, 58) | 0.30 |
| **BNP, ng/L** | 16.0 (11.0, 25.0) | 15.0 (9.0, 27.0) | 0.84 |
| **NT-proBNP, ng/L** | 17.0 (6, 31) | 20.0 (13, 39) | 0.23 |
| **Left Ventricular Hypertrophy\*** | 0 (0.0) | 2 (6.7) | 0.23 |
| **Left Atrial Enlargement** | 1 (4.8) | 7 (23.3) | 0.07 |

*SBP* systolic blood pressure, *DBP* diastolic blood pressure, *MR-proANP* mid-regional pro-atrial natriuretic peptide, *BNP* B-type natriuretic peptide, *NT-proBNP* N-terminal proB-type natriuretic peptide.
\*Using the Minnesota coding.

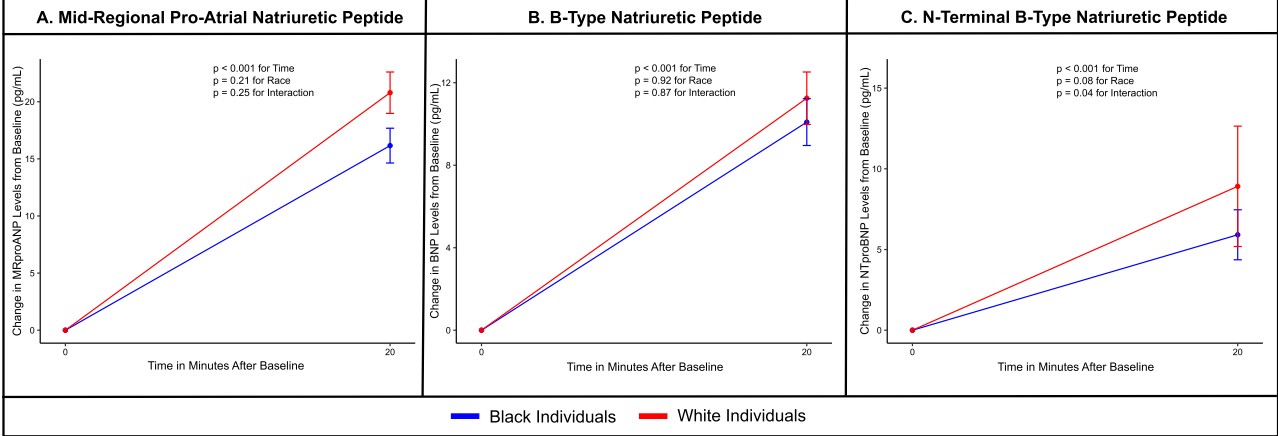

**Fig. 1 | Change in absolute natriuretic peptide concentrations with standardized exercise stratified by race.** This figure depicts the change in absolute mid-regional pro-atrial natriuretic peptide (**A**), B-type natriuretic peptide (**B**), N-terminal proB-type natriuretic peptide (**C**) concentrations with standardized exercise stratified by race. All participants underwent standardized exercise at 70% of their maximal oxygen uptake for 20 minutes. Natriuretic peptides were measured at baseline and immediately after exercise. Black ($n = 40$) and white individuals ($n = 40$) have been presented in blue and red, respectively. The values plotted depict the mean and the standard error. Linear mixed models, adjusted for age, sex, body mass index, and serum insulin concentrations, were used to assess differences in the natriuretic peptide response to standardized exercise. The two-sided *p*-values were derived from F-statistics and do not account for multiple comparisons.

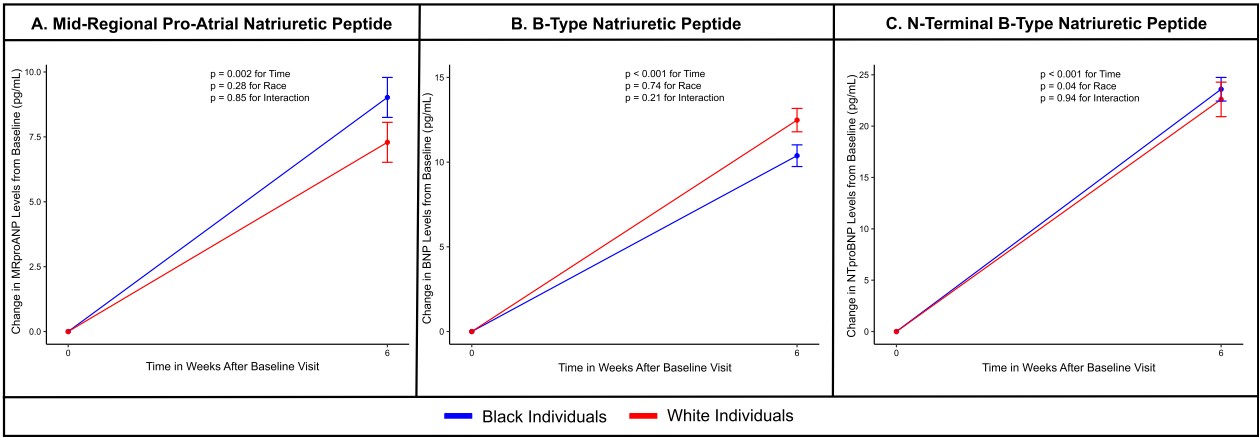

**Fig. 2 | Change in absolute natriuretic peptide concentrations with 6 weeks of metoprolol stratified by race.** This figure depicts the change in absolute mid-regional pro-atrial natriuretic peptide (**A**), B-type natriuretic peptide (**B**), N-terminal proB-type natriuretic peptide (**C**) concentrations with 6 weeks of metoprolol stratified by race. Black ($n = 40$) and white ($n = 40$) individuals have been presented in blue and red, respectively. The values plotted depict the mean and the standard error. Linear mixed models, adjusted for age, sex, body mass index, and serum insulin concentrations, were used to assess differences in the natriuretic peptide response to metoprolol at 6 weeks. The two sided $p$-value was derived from the F-statistics. The two-sided $p$-values were derived from F-statistics and do not account for multiple comparisons.

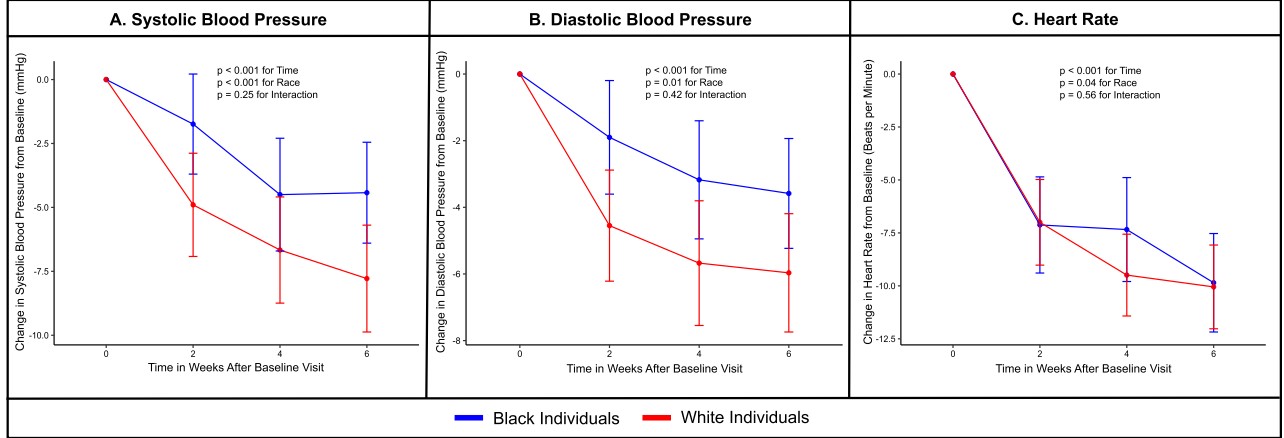

**Fig. 3 | Change in blood pressure and heart rate in response to metoprolol stratified by race.** This figure depicts the systolic blood pressure (**A**), diastolic blood pressure (**B**), and heart rate (**C**) during the medication phase of the study stratified by race. Three blood pressure and heart rate measurements were taken at baseline and every 2 weeks in each participant. The average of the three values was used. The mean and standard deviation for each measurement have been depicted. Black ($n = 40$) and white ($n = 40$) individuals have been presented in blue and red, respectively. The two-sided $p$-values were derived from F-statistics and do not account for multiple comparisons.

The observed differences in NT-proBNP and MR-proANP responses to exercise between Black and white participants could have important implications for understanding racial disparities in cardiovascular and metabolic health. It is possible that the heightened NP response in white participants may reflect differences in cardiac or vascular remodeling, or in metabolic factors that influence NP release, which could predispose individuals to distinct patterns of cardiometabolic risk. In contrast, the lower NP response in Black participants might suggest a reduced capacity for adaptive cardiovascular or metabolic responses to acute physiological stress, potentially contributing to greater susceptibility to conditions like hypertension and heart failure in the long term. However, further studies are needed to explore how these differential responses might influence long-term cardiovascular and metabolic disease development and whether targeted interventions could mitigate these risks across racial groups.

The mechanistic explanations for the increase in NP concentrations in response to exercise and metoprolol have been explored previously. Aerobic exercise is associated with physiological activation of the sympathetic nervous system[29]. This leads to increased inotropy in the heart and increases myocardial wall stress. The increase in the NPs is mediated by the increase in wall stretch with exercise[30,31]. Notably, the hemodynamic response to exercise has been shown to vary among Black and white individuals[32]. Similarly, the physiological responses to metoprolol vary by race, with the decrease in systolic blood pressure in Black individuals being approximately half of that in white individuals[33,34]. Even though the hemodynamic response to metoprolol varies by race[33,34], the NP release in response to this perturbation is similar by race. Furthermore, the decrease in plasma aldosterone concentrations, an antagonist of the NP system, was also found to be similar by race after metoprolol therapy. These findings may suggest that the sensitivity of the NP system to pharmacological perturbations is preserved in Black individuals. The current study also found that urinary sodium levels did not vary after NP augmentation with metoprolol. Although urinary sodium levels were assessed before and after NP augmentation with metoprolol, it is important to recognize that changes in urinary sodium levels may be appreciated only in the first few days after initiation, after which a steady state in sodium balance is achieved.

Previously, several studies have shown that NP concentrations increase in response to exercise. These studies demonstrate that the NP concentrations transiently increase after aerobic exercise and return to baseline concentrations within a week[31,35–40]. These studies have notable differences compared with the current study. The prior studies included individuals from a wide age range, including middle-aged individuals[31,35–40]. NP concentrations increase with age, which may reflect underlying subclinical cardiovascular disease[11,41,42]. Furthermore, these studies assessed the NP response to non-standardized loads of exercise ranging from occasional weekly exercise to ultra-marathons without accounting for salt intake[31,35–40]. NP concentrations have been shown to be affected by the amount of salt consumed through its effect on the circulating blood volume[43]. The current study utilizes a physiologically rigorous approach by using a standardized exercise regimen and comprehensively measures the change in all 3 NPs on the background of a salt-standardized diet. The NP response to metoprolol has been studied in a small sample of chronic stable heart failure patients[25]. This study showed that a 6-week regimen of metoprolol led to a ~78%, ~89%, 37%, and ~31% increase in BNP, NT-proBNP, ANP, and NT-proANP concentrations, respectively[25]. Apart from the small sample size, the prior study did not consider the role of race in the NP response. This is the first study to demonstrate the increase in NP concentrations with metoprolol in a young, healthy population without subclinical cardiovascular disease. Considering that the NP concentrations are known to increase in response to underlying cardiovascular disease and risk factors such as hypertension, these factors may preclude rigorous assessment of the NP response to perturbations. The current study enrolled young, healthy, normotensive individuals to examine the effect of a 6-week regimen of metoprolol on NP concentrations. Concordant with the prior study, NP concentrations increased with metoprolol therapy. Notably, the magnitude of increase in NT-proBNP was ~10% higher and BNP was ~20% lower in response to metoprolol in the current compared with the prior study. Our prior work on the NP system demonstrated that the suppression of the NP system using an oral glucose load was higher in self-identified white individuals than in self-identified Black individuals[7]. Contrary to prior work that evaluated the racial differences in suppression of the NP system, the current work focused on evaluating the differences in stimulation of the NP system based on self-reported race. The current study found that metoprolol was equally effective in increasing NP concentrations in Black and white individuals. Furthermore, concordant with prior literature demonstrating that beta-blockade improved insulin resistance, the current study shows that NP augmentation with metoprolol similarly improves insulin resistance among Black and white young adults[44].

The role of NPs as a biomarker and their clinical relevance indeed varies between healthy individuals and those with heart failure. In a healthy population, NP levels are typically stable, and any significant alteration may indicate an acute response to physiological stimuli, such as exercise or beta-blocker administration. The degree of responsivity of the NP system to such perturbations may serve as a surrogate of the risk of developing cardiometabolic diseases, given the regulation of cardiometabolic functions by NPs. However, in heart failure, NP concentrations are chronically elevated secondary to increased cardiac wall stress. The NPs in heart failure patients have been shown to be metabolically inactive fragments. Furthermore, NP augmentation through exercise or beta-blockade might provide insights into their role in preventing cardiometabolic diseases, particularly in patients at risk due to NP deficiency. While certain beta-blockers have been linked to metabolic disorders, the protective effects mediated by NP release could counterbalance these risks, suggesting a nuanced role for beta-blockers in managing cardiometabolic health. Recent studies, including trials such as the REDUCE-AMI[45] and ABYSS[46] trials, examining the role of beta-blocker therapy in low-risk patients with myocardial infarction, demonstrated that

discontinuing beta-blockers was not associated with a higher risk of adverse clinical outcomes in the abovementioned patient population. These findings highlight the necessity for individualizing the approach to beta-blocker therapy while taking both the cardiovascular and metabolic impacts of beta-blockers, particularly in light of their NP-augmenting effects, into account to optimize long-term health outcomes. Nonetheless, it is essential to acknowledge that the role of beta-blockers in cardiometabolic health remains complex and requires further investigation.

The standard clinical practice involves assessing NPs at resting levels, especially for patients with heart failure or suspected diastolic dysfunction. Our study emphasizes that exercise can induce physiological or pathophysiological variations in NP levels, with notable differences in NT-proBNP elevations and modest differences in MR-proANP responses between white and Black individuals. The noticeable differences in the NP response among isoforms with a longer half-life and lack of difference in BNP (NP isoform with a short half-life) may suggest that the effect of exercise on the balance between NP release and processing may vary across the NP isoforms. Hence, the time after exercise when the NP isoforms are measured may need to be individualized. Further research is needed to clarify the role of genetics and social determinants of health in influencing the beneficial response to exercise. A comprehensive understanding of these factors will be crucial for developing personalized exercise regimens, enhancing the clinical relevance of our findings.

Considering that NPs are physiologically active hormones, low NP concentrations have been shown to increase the risk of cardiometabolic diseases such as hypertension, diabetes, and hyperlipidemia[3,4,13,14]. Sustained augmentation of NPs may be beneficial in reducing the burden of cardiometabolic diseases. While the transient and sustained effects of beta-blockade on NPs have been examined among patients with dilated cardiomyopathy[47], similar studies are needed in healthy individuals to understand the physiological impact of metoprolol-induced NP augmentation. Future studies should also focus on assessing the timing of NP response to capture the differences in various circulating NP isoforms.

This study has several limitations. Considering the strict protocol followed in the study, the generalizability of the results may be limited. The study protocol required strict adherence to a 3-day standardized, mandatory fasting the night before the protocol and strict adherence to a daily dose of metoprolol. However, these measures provide an advantage in reducing inherent variation that is present in larger studies where NP concentrations are measured on a random salt background. Second, the study findings need to be replicated in populations of different ethnicities, such as Asian and Hispanic individuals and individuals from different age groups. Third, although the study was appropriately powered to detect a small-moderate effect size, the limited sample size constrained its ability to identify a very small effect size. Fourth, race is largely considered a social construct with some biological underpinnings. The current study recruited individuals based on self-reported race. However, previous studies in the Birmingham area have shown that self-identified Black individuals have ~90% African ancestry[7]. Fifth, NPs are known to have a diurnal rhythm[48]. The current study did not account for this diurnal variation to allow participants to schedule visits as per their convenience. Sixth, the study's non-randomized design could be a limitation, given that randomizing participants by race is not achievable. Seventh, due to the restriction of the study sample to young, healthy individuals, additional cardiac imaging was not obtained during the study. Last, the current study may also be limited by the reversible binding of metoprolol succinate to the beta-adrenergic receptor.

This physiologically rigorous clinical trial in healthy normotensive Black and white individuals accounting for factors affecting NP concentrations showed that the NP concentrations increased with exercise and metoprolol. Physiological interventions such as exercise may lead

to higher augmentation of plasma NP concentrations in white individuals than in Black individuals.

## Methods

### Study design, setting, and location of the clinical trial

The clinical trial was a single-center, prospective clinical trial (NCT#03070184) conducted from 2018 to 2023 at the University of Alabama at Birmingham (UAB). Participant enrollment occurred between June 16, 2017, and July 7, 2023. Healthy participants were recruited from the UAB Campus and the Birmingham metropolitan area. All participants provided informed written consent. The trial was approved by the UAB Institutional Review Board (IRB#: 170214001).

### Sample selection

This study included self-identified Black and white individuals between 18 to 40 years of age with a body mass index between 18 to 30 kg/m$^2$, seated blood pressure <140/90 mm Hg, who were willing to take the study medication and were able to exercise. Our prior work has shown that self-identified Black individuals from the Birmingham metropolitan area have a median proportion of African ancestry of ~90%. The exclusion criteria included decreased renal function (estimated glomerular filtration rate <60 mL/min/1.73 m$^2$), history of cardiovascular disease, history of diabetes, history of hypertension, anemia, pregnancy, use of hormone replacement therapy/oral contraceptives, depression, low systolic blood pressure (<100 mm Hg) or diastolic blood pressure (<60 mm Hg) at screening, low heart rate at screening (<60/min) at screening, elevated liver function tests (>3x upper normal limit), and history of smoking.

### Study protocol

Eligible participants were scheduled for a maximum exercise capacity test at the UAB Center for Exercise Medicine (UCEM) using the modified Bruce protocol administered by an exercise physiologist. Participants were monitored using a 12-lead electrocardiogram and BP measurements (Omron HEM-780; Omron Healthcare, Inc 1200 Lakeside Dr. Bannockburn, IL). Oxygen uptake and carbon dioxide production were measured using an open-circuit metabolic cart (TrueOne 2400, Parvo Medics, Sandy, UT). Achievement of maximal oxygen uptake (VO$_2$ max) was ascertained by the exercise physiologist based on standard criteria for heart rate (HR within 10 beats/min of estimated maximum), respiratory exchange ratio (RER > 1.2), and plateauing of VO$_2$/min/kg. The participants were then invited to pick up salt-standardized meals. These meals were prepared in the CRU metabolic kitchen under the supervision of the dietician. Further details are provided in the Supplementary Methods, Supplementary Note 1, and Supplementary Table 1.

The 3-day diet was followed by an exercise challenge visit conducted at UCEM by the exercise physiologist. The exercise intensity was fixed at 70% of VO$_2$ max to standardize the exercise load across participants. The exercise challenge had a 4-min warmup starting with 3 mph at zero grade for two minutes, progressing to 3 mph at a 3% grade for 1 min, followed by 3 mph at a 6% grade for one minute. Using the American College of Sports Medicine metabolic equations, treadmill grade and speed were adjusted to attain 70% VO$_2$ max. Each subject walked at 70% of their VO$_2$ max for 20 minutes. At the end of the exercise challenge, participants underwent a 4-min cool-down by walking at 2 mph at 0 grade. Before, immediately, 15 s, and 30 s after the exercise challenge, blood samples were collected to measure plasma NP concentrations. Participants were asked to provide urine samples to measure urinary sodium concentrations at the start and 30 minutes after the end of the exercise challenge.

After 3–7 days of the exercise challenge visit, the participants were scheduled for the drug initiation visit. The medication phase of the study lasted for 6 weeks, with the dose of metoprolol succinate being doubled every 2 weeks, starting from 50 mg/day. An EKG, blood pressure measurements, and a blood draw were done in the supine position during this visit. If their blood pressure measurements were within normal limits, the participants were given 50 mg of metoprolol succinate to be taken once daily for 2 weeks. The participants had follow-up visits every 2 weeks (interim visits 1 and 2 and end of study visit) at CRU, during which an EKG, blood pressure measurements, and a blood draw were done in the supine position. Compliance was assessed by pill count and the participants were asked about any possible side effects from metoprolol. If the participants were compliant with metoprolol and did not experience any adverse events from metoprolol, they were given a higher dose of metoprolol succinate to be taken once daily for 2 weeks (100 mg metoprolol succinate given at interim visit 1 and 200 mg metoprolol succinate at interim visit 2). The participants were followed up telephonically one week after the drug initiation and dose increase to monitor any adverse events to metoprolol succinate and assess compliance. Participants also collected a 24-h urine sample before the initiation of metoprolol and at 6 weeks of metoprolol to measure urinary sodium concentrations.

### Study outcomes

The primary outcome of the study was the change in the plasma NP concentrations, including NT-proBNP, BNP, and MR-proANP, in response to 6 weeks of metoprolol between young, healthy Black and white individuals. The secondary outcomes included the change in plasma NP concentrations, including NT-proBNP, BNP, and MR-proANP, immediately after exercise between Black and white individuals.

### Laboratory measurements

At each visit, blood samples were collected in EDTA and gold-top tubes. Samples were immediately processed and stored at −80 °C till the samples were used. MR-proANP (mid-regional-proANP), was measured using immunoluminometric sandwich assays (BRAHMS KRYPTOR compact plus, Hennigsdorf, Germany, minimal detectable limit: 2.1 pmol/L, interassay CV: <2% and intra-assay CV: ≤6.5%). Plasma NT-proBNP concentrations were measured using an electrochemiluminescence immunoassay (Elecsys proBNP, Roche, Indianapolis, IN; minimum detectable limit: 5 ng/L, interassay and intra-assay CV% < 2). A 2-site immunoenzymatic assay was used to measure plasma BNP (Alere Triage BNP test; minimum detectable limit: 1.0 ng/L, interassay and intra-assay CV% < 10). Serum glucose was measured using a Sirrus Stanbio (Boerne, TX) analyzer using glucose oxidase reagent (minimum detectable limit: 2 ng/L, intra-assay coefficient of variation (CV%) < 2, interassay CV% < 3). Insulin was measured using a TOSOH 900 AIA (S. San Francisco, CA; minimum detectable limit: 0.5 μU/mL, intra-assay and interassay CV% < 4). Urinary sodium was measured using Beckman Coulter AU640/AU640e (Beckman Coulter Inc., Brea, CA; intra-assay %CV < 3 and inter-assay %CV < 5). Aldosterone was measured in duplicate using the ALPCO (Salem, NH) ELISA kit, with results in pg/mL. The assay had a minimum sensitivity of 15 pg/mL, an intra-assay CV of 6.01%, and an inter-assay CV of 5.90%.

### Statistical analysis

The study was powered to detect the change in NT-proBNP in response to 6 weeks of metoprolol succinate among Black and white individuals. In a prior study evaluating NP response after 6 weeks of metoprolol (similar dose and protocol)[25], the mean ± SD relative change (measured in log units) in plasma NT-proBNP concentrations was 0.66 ± 0.45. With a prespecified sample size of 40 in each group, the study would have 80% power to detect a 28% between-group difference for the change in NT-proBNP pre- and post-metoprolol (https://clinicaltrials.gov/study/NCT03070184) (Supplementary Fig. 11). In addition, the power was assessed using effect size, calculated as the ratio of the mean difference to the standard deviation, based on the same estimates specified earlier (Supplementary Fig. 12).

All analyses were conducted on SAS 9.4 (Cary, NC). Continuous and categorical variables were compared between Black and white individuals using the Wilcoxon rank sum test and the Chi-Square test, respectively. Normality of the variables was assessed visually using Q-Q plots and using the Shapiro-Wilk test. Plasma NP concentrations were found to have a skewed distribution and were log-transformed for analysis. Considering that the NP subtypes (MR-proANP, BNP, and NT-proBNP) are not independent of each other, correction for multiple hypothesis testing was not applied. Linear mixed-effects models were used to assess the effect of metoprolol and exercise on plasma NP concentrations. These models assessed repeated NP measurement with the fixed effects of time, self-reported race, and a multiplicative interaction term of time and self-reported race. Participants were considered as random effects in the model. The models were used to calculate the least square mean values of the NPs. The above-mentioned models were further adjusted for age, sex, body mass index [BMI], and fasting insulin[7].

### Reporting summary

Further information on research design is available in the Nature Portfolio Reporting Summary linked to this article.

## Data availability

The data supporting the findings from this study are available in the Article and its Supplementary Information. Deidentified individual-level data supporting the results of the study are not publicly available to protect patient privacy. Deidentified individual-level data supporting the results of the study will be made available from the corresponding author upon reasonable request. Written access proposals should be submitted to the corresponding author at the following e-mail address: parora@uabmc.edu. The expected timeframe for responses is 6 weeks. The study protocol is available in the Supplementary Information file under Supplementary Note 1. Source data are provided with this paper.

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

## Acknowledgements
P.A. is supported by the National Heart, Lung, and Blood Institute of the National Institutes of Health (NIH) awards R01HL160982, R01HL163852, R01HL163081, and K23HL146887. The project was also supported internally by the UAB Center for Exercise Medicine. Dr. Nirav Patel is supported by the National Institutes of Health grant T32HL007457.

## Author contributions
P.A. conceptualized and designed the study. P.A., N.S.S., M.G., N.P., N.V., K.Y., T.W.B., B.G., P.L., T.J.W., and G.A. acquired, analyzed, or interpreted data. P.A., N.S.S., M.G., N.P., N.V., K.Y., J.S.D., T.W.B., B.G., P.L., T.J.W., and G.A. drafted the manuscript. All authors performed critical revisions of the manuscript. M.G. did the statistical analysis. P.A. and P.L. supervised and verified the data analysis.

## Competing interests
P.A. reports grant support from Merck Sharp & Dohme LLC and Bristol-Myers Squibb and consulting income from Bristol-Myers Squibb, which are all unrelated to this work. None of the authors had any conflicts of interest or financial disclosures to declare.

## Additional information

**Naman S. Shetty[1,2], Mokshad Gaonkar[3], Nirav Patel[3], Nehal Vekariya[3], Krishin Yerabolu[3], Jasninder S. Dhaliwal[3], Thomas W. Buford ⓘ [4,5], Barbara Gower[6], Peng Li ⓘ [7], Thomas J. Wang[8], Garima Arora[3] & Pankaj Arora ⓘ [3,9] ✉**

[1]Department of Anesthesia, Critical Care and Pain Medicine, Massachusetts General Hospital, Boston, MA, USA. [2]Harvard Medical School, Boston, MA, USA. [3]Division of Cardiovascular Disease, University of Alabama at Birmingham, Birmingham, AL, USA. [4]Department of Medicine, University of Alabama at

Birmingham, Birmingham, AL, USA. [5]Birmingham/Atlanta VA GRECC, Birmingham Veterans Affairs Medical Center, Birmingham, AL, USA. [6]Department of Nutrition Sciences, University of Alabama at Birmingham, Birmingham, AL, USA. [7]School of Nursing, University of Alabama at Birmingham, Birmingham, AL, USA. [8]Department of Medicine, University of Texas-Southwestern Medical Center, Dallas, TX, USA. [9]Section of Cardiology, Birmingham Veterans Affairs Medical Center, Birmingham, AL, USA. ✉e-mail: parora@uabmc.edu

