## [Transparent Peer Review file · Nature Communications]

Differences in Natriuretic Peptide Response in Self-Identified White and Black Individuals: A Physiological Clinical Trial

Corresponding Author: Dr Pankaj Arora

Version 0:

Reviewer comments:

Reviewer #1

(Remarks to the Author)

This is a well-conducted trial investigating the effects of exercise and a beta-blocker on plasma concentration of various natriuretic peptides (NP) according to race (Black & White) in young healthy men and women.

Abstract

The term 'levels' is ambiguous and non-scientific. "Concentration" is a better word. Please consider throughout the manuscript.

Also it is not clear whether the sampling is plasma, urine or saliva: "lower natriuretic peptide (NP) levels" would be better rendered as "lower plasma concentrations of NP".

It is disrespectful to talk about females, which makes it sound like participants might be rats; "women" and "men" are better terms.

The formulation of metoprolol should be stated (tartrate – short-acting; succinate – long-acting; although because metoprolol has highly reversible binding to the beta-receptor a better beta-blocker should have been considered (propranolol, timolol or carvedilol). This could be added to limitations.

The half-life of BNP is much shorter than NT-proBNP or MR-proANP. Long half-life may confound short-term interventions such as exercise.

General comments

The article is rather verbose in places and should be written more succinctly and avoid hyperbole.

The term 'comparable' should not be used when 'similar' is meant. Many things that are not similar can be compared (eg: an elephant and an ant).

Introduction

The authors imply that raising NP will reduce the risk of cardiometabolic diseases. I think this is very speculative and must be justified or deleted.

Methods

Suggest reducing some of the inconsequential detail. The interested reader can look at the protocol.

The dietary preparation for study visits is very important. Well done!

Statistics

All baseline data including BP etc should be reported as median with 1st/3rd quartile.

Mean/SD is inferior for descriptive data for many reasons.

It would be reasonable to changes as mean differences with 95% CI.

Results

Supplementary figure 1 is a much better illustration of the study results than is Figure 2. Log values are difficult for clinicians to interpret and are not what clinical laboratories report. I suggest swapping these. Supplementary Figure 1 suggests higher baseline MR-proANP and NT-proBNP in White people but no difference in exercise response.

Indeed, Figure 4 together with supplementary Figures 1 and 4 tell the reader all the key results of this research. Figures 1, 2 and 3 are of much less interest and can be moved to supplementary materials.

Differences in urinary sodium at baseline and 6 weeks would not be expected as participants will be in steady-state with respect to sodium balance despite changes in BNP with metoprolol. Differences would only be expected in the first few days after initiation.

Discussion

This is repetitive in places. I think shortening the discussion by a page will improve it.

I think it is far too speculative to suggest that the small rise in NP with beta-blockers will have a profound effect on CV outcomes. Large trials of beta-blockers in low-risk populations have not met with striking success. I think the authors should be much more cautious in their speculation.

Some discussion on previous trials of beta-blockers in diabetes and the effects on insulin sensitivity is warranted (eg: Bakris et al JAMA. 2004. doi: 10.1001/jama.292.18.2227).

Also, the rise in NP with withdrawal of long-term beta-blockers - suggesting this is not a transient response. (Halliday B et al. ESC Heart Fail. 2022 doi: 10.1002/ehf2.13872).

Table 1

Please report data to a sensible number of decimal places, which will usually be whole numbers (eg: BP, HR and NP) although BMI, Hb etc should be to one decimal place.

Reviewer #2

(Remarks to the Author)

This is an interesting study. I have the following comments.

In the study, the participants were recruited from the university, it's not clear if they were students or staff or mixed of students and staff.

The sample size calculation was based on NT-proBNP. However, the primary measures were also included MR-proANP and BNP. How was NT-proBNP selected for the sample size calculation? In addition, please indicate the significance level and one-tailed/two-tailed tests used in the data analysis.

As indicated that the models were additionally adjusted for age, sex, body mass index (BMI), and fasting insulin. How were the variables determined?

It looks that the trial was not randomized, which can lead to potential bias. This should be addressed in the limitation section. In addition, the lack of a control group is an important study limitation.

In the results section, p-values were mainly reported. Suggest reporting the difference with 95%CI as well.

Table 1. Noted that some data were presented as 16.9 (5.6-30.5) for example. Was "16.9 (5.6-30.5)" presented as median (IQR) or media (range)?

Reviewer #3

(Remarks to the Author)

The authors conducted a rigorous interventional study examining the circulating levels of 3 NPs after a standardized exercise protocol and a metoprolol intake program for 6 weeks. Vital measures including heart rate, blood pressure, and biomarkers levels (3 NPs) were all obtained and compared in between whites and blacks, aiming for a racial difference on NPs in response to these physiological perturbations.

Major Comments

1. The major clinical implication of this study can be further strengthened. Importantly, when we assess levels of NPs, we only obtain resting levels and rarely do exercise challenge though we know there may be variations of them (as physiological or pathophysiological responses) in between individuals when taking exercise. For HF patients or those with suspected diastolic dysfunction, we still assess resting NP levels as clinical reference;

2. One of the major is the neutral or negative findings of current study, with relatively small sample size;

3. The relative scarcity of Tables in current study should be further enriched. For example, I would appreciate if the authors also took some measures for cardiac structure (either MRI or echo-based) or functions to support some background information for the cardiac phenotypes. As we know, measure of NPs are subject to cardiac remodeling;

4. As the main hypothesis and primary endpoint is not successfully achieved, the authors might want to explore some other interesting topics. For example, African Americans may have greater extent of cardiac hypertrophy or LV filling status ? (such as invasive hemodynamic measures or E/e' by echocardiography by ergometric bicycle maneuver);

5. As it has been reported that blacks are prone to higher blood pressure and showed in general different pattern of cardiac remodeling, such as higher degree of LV hypertrophy. Did the authors also collect information about cardiac remodeling (echocardiography), for example, extent of LV hypertrophy, in response to Nt-proBNP level and compared these in between black and white and testing the potential interactions of LV mass index related to level of Nt-proBNP ?

6. Overall, the circulating levels of these NPs by interventional exercise or pharmacological use may be biologically different and can be interpreted differently. Drug exposure in current study takes weeks (up to 6 weeks), however, standard exercise program is an acute episode for the cardiac response. Therefore, NPs changes in response to drugs may be more stable (a more chronic phase balance or secretion of NPs) over time while NPs changes immediately after exercise may reflect more acute NPs alterations, either release or degradation. In Figure 2, Nt-proBNP differed substantially and significantly in response to exercise and was much higher in whites compared to black. To me, this finding is good and meaningful enough. Due to faster turn over and shorted half-life of BNP (~ 20 minutes ?), the equal alterations of BNP in response to exercise between blacks and whites may be interpreted as the balance between BNP release and turn over (degradation); however, the differences of Nt-proBNP using same exercise program and tested within 20-30 minutes may be more relevant to Nt-proBNP release (after exercise challenge) and did make a significant difference in between races. As we can see, MR-proANP elevation also showed minor differences in between races, which supported my hypothesis that variations in these 3 NPs post 20 minutes exercise may be different and partly explained by biological processing to exercise with time and may not be equal, which perhaps should be considered beyond "extent of elevations by exercise programs" simply. Therefore, the exact time point to check these NPs to obtain a more logical or precise comparisons of them are particularly important if we want to relate such NPs level differences to simply release of them in response to exercise challenge.

7. As I know, the biological degradations of BNP and Nt-proBNP are different (same with MR-proANP), and Nt-proBNP might be much longer. These biomarkers are potentially degraded by enzymes with different activity and circulation level. For Figure 2,

Minor Comments

1. Minor spelling or grammatical issues;
2. Would Supplementary Figure 2 findings (differences of spot urine sodium) explained by BNP or Nt-proBNP effect ?
3. The clinical impact and significance of NPs release by beta-blockers should be further strengthened in the Background section. I believe that many pharmacological approach can impact on NPs levels. Compared to exercise, can beta-blocker use serve as useful marker or maneuver to test NPs behavior in response to clinical pathophysiological stimuli ? If yes, what does it indicate clinically ? The observed NPs rise after either beta-blockers use or exercise may be simply related to cardiac filling stress increase and wall stress increase, respectively. Can such NPs release further translate into clinical protection of cardiometabolic diseases ? Certain beta-blockers have even been linked to metabolic disorders.

Version 1:

Reviewer comments:

Reviewer #1

(Remarks to the Author)

The authors have done an excellent job in responding to all of my comments in a positive fashion. Many thanks.

Recently, a trial of withholding beta-blockers after low-risk acute myocardial infarction or withdrawal of chronic beta-blocker therapy after myocardial infarction in low risk patients have met with slightly mixed results. It would bring the discussion up to date to mention these and comment briefly.

Yndige T et al. NEJM 2024

Silvain J et al. NEJM 2024

Reviewer #2

(Remarks to the Author)

The authors have responded well to my initial questions. I have a few comments below.

In response to the comment 2, the authors used $\text{mean} \pm \text{SD}$ as the effect size. In general, an effect size is measured as $(\text{mean difference})/(\text{SD of the difference})$. Noted that the figure shows the relationship between powers and percentage differences in means, I would suggest showing the relationship between power and effect size.

Reviewer #3

(Remarks to the Author)

The authors have made much efforts to revise the comments in a point-by-point manner from my point of view. Most comments raised have been solved adequately and yet there remains 2 issues that I am not comfortable with:

1. For figure 2 (both in current and last version), I noticed that the caption (legend) mentioned about " All participants underwent standardized exercise at 70% of their 684 maximal oxygen uptake for 20 minutes. Natriuretic peptides (NPs) were measured at baseline and 685 immediately after exercise".. However, the title and the text that referred to this Figure were mainly about "6 weeks of Metoprolol use Stratified by Race" rather than anything about exercise. The section about exercise, instead, should refer to Figure 1 ?

Please check these contents again carefully to ensure that correct legend was placed at the right place.

2. For the noticeable finding (also the key difference in this work) about "However, the NT-proBNP and MR-proANP response to exercise was found to be higher in White participants compared with Black participants".. I expect the authors to elaborate 1-2 sentences addressing the potential implications and differences on cardiac structure/function or metabolic diseases between ethnic whites or black participants, though still speculative. This would make readers re-focus on the main study objective and clinical application of the current work.

Response to Review

REVIEWER COMMENTS

Reviewer #1 (Remarks to the Author):

This is a well-conducted trial investigating the effects of exercise and a beta-blocker on plasma concentration of various natriuretic peptides (NP) according to race (Black & White) in young healthy men and women.

Abstract

1. The term ‘levels’ is ambiguous and non-scientific. “Concentration” is a better word. Please consider throughout the manuscript.

Response: We thank the Reviewer for their comments. Based on the Reviewer’s suggestion, we have replaced “levels” with “concentrations” throughout the manuscript.

2. Also it is not clear whether the sampling is plasma, urine or saliva: “lower natriuretic peptide (NP) levels” would be better rendered as “lower plasma concentrations of NP”.

Response: We appreciate the Reviewer’s feedback. All NP measurements were performed using plasma samples. As suggested by the Reviewer, we have indicated that NPs were measured in plasma samples in our revised manuscript. Furthermore, we have ensured that plasma concentrations of NPs have been mentioned across the manuscript.

3. It is disrespectful to talk about females, which makes it sound like participants might be rats; “women” and “men” are better terms.

Response: We appreciate the Reviewer’s guidance. As requested by the Reviewer, we have replaced “males” with “men” and “females” with “women” across the manuscript.

4. The formulation of metoprolol should be stated (tartrate – short-acting; succinate – long-acting; although because metoprolol has highly reversible binding to the beta-receptor a better beta-blocker should have been considered (propranolol, timolol or carvedilol). This could be added to limitations.

Response: We thank the Reviewer for their suggestion. We used the long-acting form of metoprolol, i.e., metoprolol succinate, in our study. Metoprolol was used for its cardioselective action compared with non-selective beta-blockers such as propranolol, timolol, and carvedilol. However, the highly reversible binding of metoprolol to the beta-receptor may be a limitation of the current study, as highlighted by the Reviewer. Therefore, we have noted the reversible binding of metoprolol to the beta receptor as a limitation in our revised manuscript.

Lastly, the current study may also be limited by the reversible binding of metoprolol succinate to the beta-adrenergic receptor.

5. The half-life of BNP is much shorter than NT-proBNP or MR-proANP. Long half-life may confound short-term interventions such as exercise.

Response: We are thankful for the Reviewer's feedback. The Reviewer raises a concern regarding the confounding effect of the half-life of the natriuretic peptide isoforms. In our study, we examined all the responses of all 3 natriuretic peptide isoforms, i.e., BNP, NT-proBNP, and MR-proANP. Our study found that all three isoforms showed an increase by race. Notably, the peptides with a longer half-life (NT-proBNP and MR-proANP) also showed a decrease in the 30 minutes immediately after exercise, similar to BNP. Therefore, the effect of exercise on increasing natriuretic peptide concentrations is acute and transient, irrespective of the natriuretic peptide isoform and the half-life of the natriuretic peptide isoform.

Furthermore, as pointed out by **Reviewer 3 Comment 6**, given the faster turnover and shorter half-life of BNP (approximately 20 minutes), the similar changes in BNP in response to exercise between Black and White individuals may reflect a balance between BNP release and turnover (degradation). However, the differences observed in Nt-proBNP using the same exercise program and measured within 20-30 minutes are likely more indicative of Nt-proBNP release after the exercise challenge and did show a significant difference between races. We have acknowledged this in the revised version of the manuscript.

General comments

5. The article is rather verbose in places and should be written more succinctly and avoid hyperbole.

Response: We thank the Reviewer for their suggestion. As recommended by the Reviewer, we have made our manuscript more succinct and avoided hyperbole in our revised manuscript.

6. The term 'comparable' should not be used when 'similar' is meant. Many things that are not similar can be compared (eg: an elephant and an ant).

Response: We appreciate the Reviewer's guidance. We have ensured to correctly utilize the terms 'comparable' and 'similar' in our revised manuscript.

Introduction

7. The authors imply that raising NP will reduce the risk of cardiometabolic diseases. I think this is very speculative and must be justified or deleted.

Response: We thank the Reviewer for their suggestion. The assertion that raising NP levels will reduce the risk of cardiometabolic diseases arises from several sources of evidence showing that

low NP levels are associated with the development of cardiometabolic diseases. Several studies, including our own, have shown that low NP levels are associated with higher odds of cardiometabolic diseases such as hypertension, diabetes, and hypertension.¹⁻⁶ These associations have been validated in animal models wherein NP knock out mice were shown to develop hypertension and diabetes.^{2,6} Lastly, the causal role of low NPs in the development of diabetes has been established in Mendelian randomization studies.⁴

As suggested by the Reviewer, we have added these lines of evidence to justify that raising NP will reduce the risk of cardiometabolic diseases.

Page 5, Line 89

The role of low NP concentrations in the development of cardiometabolic diseases has been delineated through multiple lines of evidence, including epidemiological, animal, and genetic studies.¹⁻⁶

Methods

8. Suggest reducing some of the inconsequential detail. The interested reader can look at the protocol.

Response: We appreciate the Reviewer's suggestion. We have shortened the methods section to remove less important details of the study and have directed readers to the protocol for detailed information on the study methods.

9. The dietary preparation for study visits is very important. Well done!

Response: We thank the Reviewer for their recognizing the rigor of our study methodology.

Statistics

10. All baseline data including BP etc should be reported as median with 1st/3rd quartile. Mean/SD is inferior for descriptive data for many reasons.

Response: We thank the Reviewer for their suggestion. Based on the Reviewer's suggestion, we have now updated the baseline table wherein we have used the median along with the IQR to summarize continuous variables. The updated baseline table has been shown in **Reviewer Table 1**.

Reviewer Table 1: Baseline Characteristics Stratified by Race

	Black Participants (n=40)	White Participants (n=40)	p-value
Age, years	27 (22, 32)	25 (20, 30)	0.10
Women, n (%)	21 (52.5%)	19 (47.5%)	0.66
Body Mass Index, kg/m²	25.1 (22.4, 27.7)	24.1 (22.4, 26.8)	0.58
SBP, mmHg	109 (105, 116)	108 (102, 116)	0.55
DBP, mmHg	67 (64, 74)	70 (63, 75)	0.48
Heart rate, beats per minute	69 (63, 77)	70 (62.7, 79)	0.74
Fasting Glucose, mg/dL	83.0 (80.0, 89.0)	86.5 (79.0, 90.0)	0.54
Hemoglobin, g/dL	13.3 (12.5, 14.1)	14.1 (12.6, 14.8)	0.08
Insulin, mg/dL	6.5 (3.9, 9.0)	5.9 (3.7, 7.9)	0.25
Creatinine, mg/dL	1.0 (0.8, 1.1)	0.9 (0.7, 1.0)	0.11
AST U/L	17.0 (15.0, 25.0)	17.0 (14.5, 21.0)	0.61
ALT U/L	14.0 (11.0, 18.0)	14.5 (12.0, 20.5)	0.76
MR-proANP, pmol/L	43 (31, 56)	47 (35, 58)	0.30
BNP, ng/L	16.0 (11.0, 25.0)	15.0 (9.0, 27.0)	0.84
NT-proBNP, ng/L	17.0 (6, 31)	20.0 (13, 39)	0.23

11. It would be reasonable to changes as mean differences with 95% CI.

Response: We thank the Reviewer for their suggestion. As requested by the Reviewer, we have updated the changes in natriuretic peptide levels in our study as mean differences with 95% confidence interval.

Results

12. Supplementary figure 1 is a much better illustration of the study results than is Figure 2. Log values are difficult for clinicians to interpret and are not what clinical laboratories report. I suggest swapping these. Supplementary Figure 1 suggests higher baseline MR-proANP and NT-proBNP in White people but no difference in exercise response.

Response: We appreciate the Reviewer's comment. We agree with the Reviewer that interpreting the absolute natriuretic peptide values is more intuitive compared with the log-transformed natriuretic peptide values. As recommended by the Reviewer, we have included the absolute change in natriuretic peptides with the study interventions in the main manuscript and have moved the change in log-transformed natriuretic peptide levels to the **Supplementary Materials** in our submission.

13. Indeed, Figure 4 together with supplementary Figures 1 and 4 tell the reader all the key results of this research. Figures 1, 2 and 3 are of much less interest and can be moved to supplementary materials.

Response: We thank the Reviewer for their suggestion. Based on the Reviewer's suggestion, we have now included **Figure 4, Supplementary Figure 1, and Supplementary Figure 4** to the main manuscript and **Figures 1-3** to the **Supplementary Materials**.

14. Differences in urinary sodium at baseline and 6 weeks would not be expected as participants will be in steady-state with respect to sodium balance despite changes in BNP with metoprolol. Differences would only be expected in the first few days after initiation.

Response: We appreciate the Reviewer's suggestion. The measurement of 24-hour urinary sodium showed that urinary sodium excretion did not change significantly with natriuretic peptide augmentation using metoprolol in White or Black individuals. The Reviewer astutely highlights that a change in urinary sodium excretion would not be expected over the 6-week period due to the achievement of a steady state in sodium balance because of the rigorous dietary protocol. We have included this explanation as a possible cause of our study findings in our revised manuscript.

Page 17, Line 328

The current study also found that urinary sodium levels did not vary after NP augmentation with metoprolol. Although urinary sodium levels were assessed before and after NP augmentation with metoprolol, it is important to recognize that changes in urinary sodium levels may be appreciated only in the first few days after initiation, after which a steady state in sodium balance is achieved.

Discussion

15. This is repetitive in places. I think shortening the discussion by a page will improve it.

Response: We thank the Reviewer for their comment. As suggested by the Reviewer, we have shortened the manuscript by nearly a page to improve the readability of the manuscript.

16. I think it is far too speculative to suggest that the small rise in NP with beta-blockers will have a profound effect on CV outcomes. Large trials of beta-blockers in low-risk populations have not met with striking success. I think the authors should be much more cautious in their speculation.

Response: We thank the Reviewer for their suggestion. We agree with the Reviewer that the beta-blocker mediated NP increase is unlikely to have any profound effect on CV outcomes. As suggested by the Reviewer, we have exercised additional caution while describing the probable clinical implication of augmenting the natriuretic peptide system using beta-blockers.

17. Some discussion on previous trials of beta-blockers in diabetes and the effects on insulin sensitivity is warranted (eg: Bakris et al JAMA. 2004. doi: 10.1001/jama.292.18.2227).

Response: We thank the Reviewer for their comment. We read the manuscript provided by the Reviewer with great interest. Based on the Reviewer's suggestion, we examined the change in insulin sensitivity as quantified by the HOMA-IR. We found that natriuretic peptide augmentation using metoprolol was associated with a decrease in HOMA-IR, which was similar by race. (**Reveiw Figure 1**) Therefore, insulin sensitivity increased with metoprolol augmentation concordant with prior data. We have also supplemented our manuscript with this analysis (**Supplementary Figure 7**) and the discussion section with the prior beta-blocker trials examining insulin sensitivity.

Reviewer Figure 1: Change in Least Square Means of HOMA-IR in Response to 6 Weeks of Metoprolol Stratified by Race

Page 18, Line 362

Furthermore, concordant with prior literature demonstrating that beta-blockade improved insulin resistance, the current study shows that NP augmentation with metoprolol similarly improves insulin resistance among Black and White young adults.⁷

18. Also, the rise in NP with withdrawal of long-term beta-blockers - suggesting this is not a transient response. (Halliday B et al. ESC Heart Fail. 2022 doi: 10.1002/ehf2.13872).

Response: We thank the Reviewer for their suggestion. We read the reference that the Reviewer has shared with us and have added that in the discussion section. The manuscript shared by the Reviewer demonstrates that NT-proBNP concentrations fell transiently after withdrawal from therapy but rose at 6 months after withdrawal of therapy among patients with dilated cardiomyopathy. The transient decrease in NT-proBNP concentrations may be secondary to the action of beta-blockers (negative inotropy and increased lusitropy). However, the increase in NT-proBNP concentrations at 6 months may be attributed to clinical deterioration of heart failure. Based on the manuscript shared by the Reviewer, we have acknowledged the need to examine the sustained effect of beta-blockade in healthy individuals.

While the transient and sustained effects of beta-blockade on NPs have been examined among patients with dilated cardiomyopathy,⁸ similar studies are needed in healthy individuals to understand the physiological impact of metoprolol-induced NP augmentation.

Table 1

19. Please report data to a sensible number of decimal places, which will usually be whole numbers (eg: BP, HR and NP) although BMI, Hb etc should be to one decimal place.

Response: We thank the Reviewer for their suggestion. Based on Reviewer’s suggestion we have now reported traits like systolic blood pressure, diastolic blood pressure, heart rate, MR-proANP, BNP, and NT-proBNP as whole numbers and other continuous traits rounded to 1 decimal. The updated table is attached above (**Reviewer Table 2**)

Reviewer Table 2: Baseline Characteristics Stratified by Race

	Black Participants (n=40)	White Participants (n=40)	p-value
Age, years	27 (22, 32)	25 (20, 30)	0.10
Women, n (%)	21 (52.5%)	19 (47.5%)	0.66
Body Mass Index, kg/m²	25.1 (22.4, 27.7)	24.1 (22.4, 26.8)	0.58
SBP, mmHg	109 (105, 116)	108 (102, 116)	0.55
DBP, mmHg	67 (64, 74)	70 (63, 75)	0.48
Heart rate, beats per minute	69 (63, 77)	70 (62.7, 79)	0.74
Fasting Glucose, mg/dL	83.0 (80.0, 89.0)	86.5 (79.0, 90.0)	0.54
Hemoglobin, g/dL	13.3 (12.5, 14.1)	14.1 (12.6, 14.8)	0.08
Insulin, mg/dL	6.5 (3.9, 9.0)	5.9 (3.7, 7.9)	0.25
Creatinine, mg/dL	1.0 (0.8, 1.1)	0.9 (0.7, 1.0)	0.11
AST U/L	17.0 (15.0, 25.0)	17.0 (14.5, 21.0)	0.61
ALT U/L	14.0 (11.0, 18.0)	14.5 (12.0, 20.5)	0.76

MR-proANP, pmol/L	43 (31, 56)	47 (35, 58)	0.30
BNP, ng/L	16.0 (11.0, 25.0)	15.0 (9.0, 27.0)	0.84
NT-proBNP, ng/L	17.0 (6, 31)	20.0 (13, 39)	0.23

Reviewer #2 (Remarks to the Author):

This is an interesting study. I have the following comments.

Response: We thank the Reviewer for their thorough review of our manuscript. We have attempted to address the concerns raised by the Reviewer in our revised manuscript. We believe that the Reviewer's comments have significantly improved the quality of our manuscript.

1. In the study, the participants were recruited from the university, it's not clear if they were students or staff or mixed of students and staff.

Response: We thank the Reviewer for their feedback. As requested by the Reviewer,. To clarify, the clinical trial (NCT# 03070184) was a single-center, prospective clinical trial conducted from 2018 to 2023 at the University of Alabama at Birmingham (UAB). Healthy participants were recruited from both the UAB Campus and the Birmingham metropolitan area. Study advertisements were disseminated via recruitment flyers posted in community locations, such as churches, libraries, and at the University of Alabama at Birmingham. Our recruitment process targeted both students and staff members at UAB, as well as members of the broader Birmingham community. All participants who met the rigorous inclusion criteria of our study, irrespective of the source of recruitment, were included in the study.

2. The sample size calculation was based on NT-proBNP. However, the primary measures were also included MR-proANP and BNP. How was NT-proBNP selected for the sample size calculation? In addition, please indicate the significance level and one-tailed/two-tailed tests used in the data analysis.

Response: We thank the Reviewer for their feedback. The effect size of NP response after 6 weeks of metoprolol was taken to be 0.66 ± 0.45 (Mean \pm SD relative change measured in log-units) based on a prior study. At the time of drafting the proposal, 5 subjects had already completed the study, with their NPs measured and NT-proBNP showing a 110% increase after 6 weeks of metoprolol. Based on the power curve provided below (**Reviewer Figure 2**), we have strong statistical power to detect relatively small to medium differences (28%) in the changes in NT-proBNP levels between Black and White individuals. Even with an attrition rate of 20% (leaving 32 participants in each group due to loss to follow-up or withdrawal), the detectable difference only slightly changes from 28% to 32%

Reviewer Figure 2: Power Curve for detecting mean fold-change in NTproBNP levels between African-Americans and Whites from pre to post-6 weeks of metoprolol

3. As indicated that the models were additionally adjusted for age, sex, body mass index (BMI), and fasting insulin. How were the variables determined?

Response: We appreciate the Reviewer’s feedback. The pre-determined covariates for adjustment were selected based on variables known to influence natriuretic peptide levels.⁹ Several studies, including our own, have demonstrated that natriuretic peptide levels increase with age, decrease with increasing BMI and insulin levels, and are higher in women. To account for the variation in natriuretic levels due to these factors, we adjusted our statistical models with these variables.

4. It looks that the trial was not randomized, which can lead to potential bias. This should be addressed in the limitation section. In addition, the lack of a control group is an important study limitation.

Response: We thank the Reviewer for their feedback. The Reviewer correctly observes the non-randomized nature of our clinical trial and the lack of a control group. Prior evidence has demonstrated that our study interventions (exercise and metoprolol) increase natriuretic peptide levels. Based on this background, our study focused on evaluating if the increase in natriuretic peptide with the study interventions differed by race. This hypothesis was based on the observation that baseline natriuretic peptide levels vary by race and Black race is a state of natriuretic peptide deficiency.^{1,9-11}

Furthermore, the strict inclusion criteria and standardized dietary interventions add rigor to the methodology of our trial. Nonetheless, we have acknowledged that the trial was limited by the lack of randomization since randomization by race is not feasible. The whites with normal baseline natriuretic peptide levels served as the control group.

Page 21, Line 413

Sixth, the study's non-randomized design could be a limitation, given that randomizing participants by race is not achievable.

5. In the results section, p-values were mainly reported. Suggest reporting the difference with 95%CI as well.

Response: We thank the Reviewer for their comment. We have now added the difference values along with their 95% confidence interval.

6. Table 1. Noted that some data were presented as 16.9 (5.6-30.5) for example. Was “16.9 (5.6-30.5)” presented as median (IQR) or median (range)?

Response: We thank the Reviewer for their suggestion. As suggested by **Reviewers 1 and 2**, we have presented continuous data as median along with 1st and 3rd quartiles in our revised manuscript. Additionally, we have clarified the values presented in our manuscript in the statistical analysis section.

Reviewer #3 (Remarks to the Author):

The authors conducted a rigorous interventional study examining the circulating levels of 3 NPs after a standardized exercise protocol and a metoprolol intake program for 6 weeks. Vital measures including heart rate, blood pressure, and biomarkers levels (3 NPs) were all obtained and compared in between whites and blacks, aiming for a racial difference on NPs in response to these physiological perturbations.

Major Comments

1. The major clinical implication of this study can be further strengthened. Importantly, when we assess levels of NPs, we only obtain resting levels and rarely do exercise challenge though we know there may be variations of them (as physiological or pathophysiological responses) in between individuals when taking exercise. For HF patients or those with suspected diastolic dysfunction, we still assess resting NP levels as clinical reference.

Response: We appreciate the Reviewer's comment. Our study was based on the background that higher natriuretic peptide levels in healthy individuals have been shown to improve cardiometabolic health.^{1-6,12} Several lines of evidence support that high natriuretic peptide levels in healthy individuals decrease the risk of diabetes, hypertension, and hypercholesterolemia.^{1-6,12} Several studies, including our own, have shown that healthy Black individuals have lower natriuretic peptide levels compared with healthy White individuals.^{1,9,13-15} The relatively lower natriuretic peptide levels in Black individuals may partially explain the higher burden of cardiometabolic diseases in Black individuals. Given the cardiometabolic benefits of natriuretic peptide augmentation, we aimed to assess if transient and chronic interventions increased natriuretic peptide similarly in Black and White individuals. Our study supports that interventions aimed at augmenting natriuretic peptide levels were similarly effective in Black and White individuals. Therefore, these interventions may be effectively implemented irrespective of race to increase natriuretic peptide and improve cardiometabolic health. Given the transient improvement in natriuretic peptide levels with exercise, a regular exercise plan may be used to increase natriuretic peptide levels. Otherwise, pharmacological augmentation with metoprolol may be utilized to elicit a sustained improvement in natriuretic peptide levels.

Page 19, Line 379

The standard clinical practice involves assessing NPs at resting levels, especially for patients with heart failure or suspected diastolic dysfunction. Our study emphasizes that exercise can induce physiological or pathophysiological variations in NP levels, with notable differences in NT-proBNP elevations and minor differences in MR-proANP responses between White and Black individuals. Further research is needed to clarify the role of genetics and social determinants of health in influencing the beneficial response to exercise. A comprehensive understanding of these factors will be crucial for developing personalized exercise regimens, enhancing the clinical relevance of our findings.

2. One of the major is the neutral or negative findings of current study, with relatively small sample size.

Response: We appreciate the Reviewer's guidance. Our study sample size was derived based on calculations that afforded us high power to detect a small to medium effect size, as outlined in **Reviewer 2 Comment 2**. Furthermore, our stringent inclusion and exclusion criteria and rigorous methodology limited the amount of confounders in our study. Nonetheless, we acknowledge that our study may have been limited in detecting a small effect size due to the restricted sample size. We have updated the limitation section of our manuscript accordingly.

Page 20, Line 407

Thirdly, although the study was appropriately powered to detect a small-moderate effect size, the limited sample size constrained its ability to identify a very small effect size.

3. The relative scarcity of Tables in current study should be further enriched. For example, I would appreciate if the authors also took some measures for cardiac structure (either MRI or echo-based) or functions to support some background information for the cardiac phenotypes. As we know, measure of NPs are subject to cardiac remodeling.

Response: We appreciate the Reviewer's suggestion. In our study, we ensured that a healthy sample was selected. We used strict inclusion and exclusion criteria and restricted the age group to young adults (18-40 years) while recruiting study participants to exclude individuals with prevalent diseases. Furthermore, each participant underwent a physical examination and a series of blood tests to ensure that they were healthy and met the criteria for our study. Lastly, participants also underwent electrocardiography to ensure that they did not have any underlying structural abnormalities or rhythm disorders. **Reviewer Table 3** shows that the cardiac structure (i.w. left ventricular hypertrophy and left atrial enlargement) based on electrocardiography did not vary between the Black and White participants in our study. We have supplemented our baseline table with this data. Due to the stringent criteria implemented and the recruitment of young and healthy individuals in our study, we did not obtain additional imaging such as echocardiography to examine cardiac structure and function. We have acknowledged the lack of cardiac imaging in the limitation section of our manuscript.

Reviewer Table 3: Baseline Cardiac Structure Stratified by Race

Characteristics	Overall (n=51)	White Participants (n=21)	Black Participants (n=30)	p *
Left Ventricular Hypertrophy (LVH) based on Minnesota coding	2 (3.9)	0 (0.0)	2 (6.7)	0.23
Left Atrial Enlargement (LAE)	8 (15.7)	1 (4.8)	7 (23.3)	0.07

*p-value corresponds to the Chi-square test of Independence comparing the differences in proportions between White and Black participants.

Page 21, Line 415

Seventh, due to the restriction of the study sample to young, healthy individuals, additional cardiac imaging was not obtained during the study.

Additionally, to further enhance the data in our manuscript, we also examined the racial differences in the aldosterone response to 6 weeks of metoprolol. Aldosterone was measured in duplicate using the ALPCO (Salem, NH) ELISA kit, with results in pg/mL. The assay had a minimum sensitivity of 15 pg/mL, an intra-assay CV of 6.01%, and an inter-assay CV of 5.90%. The decrease in aldosterone concentrations was similar in both Black and White participants. **(Reviewer Figure 3)** Alongside the increase in NPs, the decrease in aldosterone concentrations suggests a physiological antagonism between the NP and RAAS systems, as previously described.¹⁶ Although aldosterone levels are not a direct measure of cardiac structure or function, they can be considered a surrogate marker due to their role in the RAAS, which is closely related to cardiac remodeling processes. The analysis on plasma aldosterone concentrations has been included in the **Supplementary Materials. (Supplementary Figure 8)**

Page 10, Line 191

Aldosterone was measured in duplicate using the ALPCO (Salem, NH) ELISA kit, with results in pg/mL. The assay had a minimum sensitivity of 15 pg/mL, an intra-assay CV of 6.01%, and an inter-assay CV of 5.90%.

Page 17, Line 325

Furthermore, the decrease in plasma aldosterone concentrations, an antagonist of the NP system, was also found to be similar by race after metoprolol therapy.

Reviewer Figure 3: Changes in Plasma Aldosterone Concentrations Following 6 Weeks of Metoprolol Stratified by Race

4. As the main hypothesis and primary endpoint is not successfully achieved, the authors might want to explore some other interesting topics. For example, African Americans may have greater extent of cardiac hypertrophy or LV filling status? (such as invasive hemodynamic measures or E/e' by echocardiography by urgometric bicycle maneuver).

Response: We appreciate the Reviewer's guidance. Although the Reviewer notes that the primary endpoint was not met in our study, we believe that the null results are also salient. Our study supports that natriuretic peptide augmentation using exercise and metoprolol was associated with a similar increase in natriuretic peptide levels in Black and White individuals. Therefore, interventions such as exercise and metoprolol are equally beneficial in improving natriuretic peptide levels in Black and White individuals. Considering that Black individuals have a higher burden of cardiometabolic diseases, natriuretic peptide augmentation in Black individuals may be beneficial in improving cardiometabolic health in Black individuals. We performed electrocardiography in our participants to rule out any abnormalities in cardiac structure and conduction disorders. We found that the presence of left atrial enlargement and left ventricular hypertrophy did not vary between the Black and White participants in our study. However, we agree with the Reviewer that including additional measures such as echocardiography would have been beneficial. We have acknowledged the lack of

echocardiography data in the limitation section of our manuscript.

Reviewer Table 4: Baseline Cardiac Structure Stratified by Race

Characteristics	Overall (n=51)	White Participants (n=21)	Black Participants (n=30)	p *
Left Ventricular Hypertrophy (LVH) based on Minnesota coding	2 (3.9)	0 (0.0)	2 (6.7)	0.23
Left Atrial Enlargement (LAE)	8 (15.7)	1 (4.8)	7 (23.3)	0.07

*p-value corresponds to the Chi-square test of Independence comparing the differences in proportions between White and Black participants.

Page 21, Line 415

Seventh, due to the restriction of the study sample to young, healthy individuals, additional cardiac imaging was not obtained during the study.

5. As it has been reported that blacks are prone to higher blood pressure and showed in general different pattern of cardiac remodeling, such as higher degree of LV hypertrophy. Did the authors also collect information about cardiac remodeling (echocardiography), for example, extent of LV hypertrophy, in response to Nt-proBNP level and compared these in between black and white and testing the potential interactions of LV mass index related to level of Nt-proBNP ?

Response: We appreciate the Reviewer’s comment. We agree with the Reviewer that several population-level studies, including our own, have demonstrated that Black individuals have higher blood pressure than White individuals, which may contribute to the racial differences in the pattern of cardiac remodeling. To circumvent any confounding of our study results due to structural cardiac abnormalities, we implemented strict criteria to identify a healthy subset of individuals for our study. **Table 1** depicts that not only the blood pressure but also other factors that may affect natriuretic peptide levels, such as body mass index, fasting glucose levels, fasting insulin levels, and creatinine levels, were in the normal range and similar in our Black and White subsets. However, we agree with the Reviewer that additional cardiac imaging would have enhanced our study. We have acknowledged the lack of echocardiography in our revised manuscript.

Page 21, Line 415

Seventh, due to the restriction of the study sample to young, healthy individuals, additional cardiac imaging was not obtained during the study.

6. Overall, the circulating levels of these NPs by interventional exercise or pharmacological use may be biologically different and can be interpreted differently. Drug exposure in current study takes weeks (up to 6 weeks), however, standard exercise program is an acute episode for the cardiac response. Therefore, NPs changes in response to drugs may be more stable (a more chronic phase balance or secretion of NPs) over time while NPs changes immediately after exercise may reflect more acute NPs alterations, either release or degradation. In Figure 2, Nt-proBNP differed substantially and significantly in response to exercise and was much higher in whites compared to black. To me, this finding is good and meaningful enough. Due to faster turn over and shorted half-life of BNP (~ 20 minutes ?), the equal alterations of BNP in response to exercise between blacks and whites may be interpreted as the balance between BNP release and turn over (degradation); however, the differences of Nt-proBNP using same exercise program and tested within 20-30 minutes may be more relevant to Nt-proBNP release (after exercise challenge) and did make a significant difference in between races. As we can see, MR-proANP elevation also showed minor differences in between races, which supported my hypothesis that variations in these 3 NPs post 20 minutes exercise may be different and partly explained by biological processing to exercise with time and may not be equal, which perhaps should be considered beyond “extent of elevations by exercise programs” simply. Therefore, the exact time point to check these NPs to obtain a more logical or precise comparisons of them are particularly important if we want to relate such NPs level differences to simply release of them in response to exercise challenge.

Response: We thank the Reviewer for their suggestion. We appreciate the Reviewer’s attention to the role of nature (exercise vs. pharmacological augmentation) and duration (transient with exercise and chronic with metoprolol) to the biological differences in NP responses. We agree that the chronic changes in NP concentrations associated with metoprolol use over several weeks may be distinct from the transient acute responses observed immediately after exercise. Based on the Reviewer’s comment, we have supplemented the discussion section in the manuscript to highlight these differences.

Page 16, Line 301

Standardized exercise was associated with a significantly higher increase in plasma NT-proBNP concentrations in White participants (23%) compared with Black participants (11%). Although not statistically, the increase in MR-proANP immediately after exercise was also higher in White participants (43%) than in Black participants (35%).

Page 19, Line 379

The standard clinical practice involves assessing NPs at resting levels, especially for patients with heart failure or suspected diastolic dysfunction. Our study emphasizes that exercise can induce physiological or pathophysiological variations in NP levels, with notable differences in NT-proBNP elevations and modest differences in MR-proANP responses between White and Black individuals. The noticeable racial differences in the NP response among isoforms with a longer half-life and lack of difference in BNP (NP isoform with a short half-life) may suggest that the effect of exercise on the balance

between NP release and processing may vary across the NP isoforms. Hence, the time after exercise when the NP isoforms are measured may need to be individualized. Further research is needed to clarify the role of genetics and social determinants of health in influencing the beneficial response to exercise. A comprehensive understanding of these factors will be crucial for developing personalized exercise regimens, enhancing the clinical relevance of our findings.

7. As I know, the biological degradations of BNP and Nt-proBNP are different (same with MR-proANP), and Nt-proBNP might be much longer. These biomarkers are potentially degraded by enzymes with different activity and circulation level. For Figure 2.

Response: We thank the Reviewer for their comment. The Reviewer highlights that the NP isoforms are degraded by different mechanisms that may contribute to the study findings. ANP and BNP are known to be mainly degraded by neprilysin while the mechanism of NT-proBNP excretion occurs renally but the mechanism is not completely known. As per the Reviewer's guidance, we have highlighted that the differences in the degradation of the NP isoforms may contribute to the study results in the discussion section of our revised manuscript.

Minor Comments

8. Minor spelling or grammatical issues.

Response: We appreciate the Reviewer's guidance. We have thoroughly edited the manuscript to correct all grammatical issues.

9. Would Supplementary Figure 2 findings (differences of spot urine sodium) explained by BNP or Nt-proBNP effect ?

Response: We appreciate the Reviewer's comment. Biologically, BNP is an active hormone, while NT-proBNP is an inactive fragment. Therefore, any biological effects such as natriuresis could be attributed to the action of BNP.

However, as highlighted by **Reviewer 1**, chronic natriuretic peptide augmentation is unlikely to affect urinary sodium excretion due to the achievement of a steady state in sodium balance over the study. Our study demonstrates that urinary sodium excretion after chronic natriuretic peptide augmentation with metoprolol did not change from baseline in Black and White individuals.

10. The clinical impact and significance of NPs release by beta-blockers should be further strengthened in the Background section. I believe that many pharmacological approach can impact on NPs levels. Compared to exercise, can beta-blocker use serve as useful marker or maneuver to test NPs behavior in response to clinical pathophysiological stimuli ? If yes, what does it indicate clinically ? The observed NPs rise after either beta-blockers use or exercise may be simply related to cardiac filling stress increase and wall stress

increase, respectively. Can such NPs release further translate into clinical protection of cardiometabolic diseases ? Certain beta-blockers have even been linked to metabolic disorders.

Response: We thank the Reviewer for their suggestion. The Reviewer astutely notes that several medications may impact NP concentrations. We considered these alternate medications while designing our study. Although NP analogs, such as nesiritide, are available, the intravenous route of administration and short half-life of the medication limited its utility for our study. Most other medications have been examined in the context of decreasing NP levels. Metoprolol was selected for our study due to its safety record and prior data supporting that metoprolol effectively increases NP levels.

The Reviewer raises a pertinent concern regarding the responsivity of the NP system to stimuli. It could be hypothesized that individuals with a responsive NP system i.e. a higher increase in NP levels with a perturbation, are likely to have a lower burden of cardiometabolic disease compared with individuals with a less responsive NP system. Our study provides a background for future investigations examining the role of the NP response in influencing the cardiometabolic disease burden and development of cardiovascular disease by showing that the NP response to perturbations does not vary by race among healthy individuals.

The benefit of NP augmentation, either through exercise or pharmacological augmentation using metoprolol, is supported by several studies demonstrating that high NP concentrations are associated with a lower burden of cardiometabolic disease. Using nationwide population-level data, we have shown that individuals with higher NP levels have lower odds of hypertension, diabetes, and hypertension.¹ However, we acknowledge that additional studies are required to examine the role of NP augmentation in improving cardiometabolic health and assessing the risk-benefit ratio of pharmacological augmentation of the NP system.

Page 19, Line 366

The role of NPs as a biomarker and their clinical relevance indeed varies between healthy individuals and those with heart failure. In a healthy population, NP levels are typically stable, and any significant alteration may indicate an acute response to physiological stimuli, such as exercise or beta-blocker administration. The degree of responsivity of the NP system to such perturbations may serve as a surrogate of the risk of developing cardiometabolic diseases, given the regulation of cardiometabolic functions by NPs. However, in heart failure, NP concentrations are chronically elevated secondary to increased cardiac wall stress. The NPs in heart failure patients have been shown to be metabolically inactive fragments. Furthermore, NP augmentation through exercise or beta-blockade might provide insights into their role in preventing cardiometabolic diseases, particularly in patients at risk due to NP deficiency. While certain beta-blockers have been linked to metabolic disorders, the protective effects mediated by NP release could counterbalance these risks, suggesting a nuanced role for beta-blockers in managing cardiometabolic health.

References

- 1 Shetty, N. S. *et al.* Natriuretic Peptide Normative Levels and Deficiency: The National Health and Nutrition Examination Survey. *JACC Heart Fail* **12**, 50-63, doi:10.1016/j.jchf.2023.07.018 (2024).
- 2 Holditch, S. J. *et al.* B-Type Natriuretic Peptide Deletion Leads to Progressive Hypertension, Associated Organ Damage, and Reduced Survival: Novel Model for Human Hypertension. *Hypertension* **66**, 199-210, doi:10.1161/HYPERTENSIONAHA.115.05610 (2015).
- 3 Echouffo-Tcheugui, J. B. *et al.* Insulin Resistance and N-Terminal Pro-B-Type Natriuretic Peptide Among Healthy Adults. *JAMA Cardiol* **8**, 989-995, doi:10.1001/jamacardio.2023.2758 (2023).
- 4 Pfister, R. *et al.* Mendelian randomization study of B-type natriuretic peptide and type 2 diabetes: evidence of causal association from population studies. *PLoS Med* **8**, e1001112, doi:10.1371/journal.pmed.1001112 (2011).
- 5 Newton-Cheh, C. *et al.* Association of common variants in NPPA and NPPB with circulating natriuretic peptides and blood pressure. *Nat Genet* **41**, 348-353, doi:10.1038/ng.328 (2009).
- 6 Coue, M. *et al.* Defective Natriuretic Peptide Receptor Signaling in Skeletal Muscle Links Obesity to Type 2 Diabetes. *Diabetes* **64**, 4033-4045, doi:10.2337/db15-0305 (2015).
- 7 Bakris, G. L. *et al.* Metabolic effects of carvedilol vs metoprolol in patients with type 2 diabetes mellitus and hypertension: a randomized controlled trial. *JAMA* **292**, 2227-2236, doi:10.1001/jama.292.18.2227 (2004).
- 8 Halliday, B. P. *et al.* Changes in clinical and imaging variables during withdrawal of heart failure therapy in recovered dilated cardiomyopathy. *ESC Heart Fail* **9**, 1616-1624, doi:10.1002/ehf2.13872 (2022).
- 9 Patel, N. *et al.* Race, Natriuretic Peptides, and High-Carbohydrate Challenge: A Clinical Trial. *Circ Res* **125**, 957-968, doi:10.1161/CIRCRESAHA.119.315026 (2019).
- 10 Bajaj, N. S. *et al.* Racial Differences in Plasma Levels of N-Terminal Pro-B-Type Natriuretic Peptide and Outcomes: The Reasons for Geographic and Racial Differences in Stroke (REGARDS) Study. *JAMA Cardiol* **3**, 11-17, doi:10.1001/jamacardio.2017.4207 (2018).
- 11 Patel, N. *et al.* Race-based demographic, anthropometric and clinical correlates of N-terminal-pro B-type natriuretic peptide. *Int J Cardiol* **286**, 145-151, doi:10.1016/j.ijcard.2019.02.034 (2019).
- 12 Newton-Cheh, C. *et al.* Genome-wide association study identifies eight loci associated with blood pressure. *Nat Genet* **41**, 666-676, doi:10.1038/ng.361 (2009).
- 13 Gupta, D. K. *et al.* Racial differences in circulating natriuretic peptide levels: the atherosclerosis risk in communities study. *J Am Heart Assoc* **4**, doi:10.1161/JAHA.115.001831 (2015).
- 14 Gupta, D. K., de Lemos, J. A., Ayers, C. R., Berry, J. D. & Wang, T. J. Racial Differences in Natriuretic Peptide Levels: The Dallas Heart Study. *JACC Heart Fail* **3**, 513-519, doi:10.1016/j.jchf.2015.02.008 (2015).

- 15 Gupta, D. K. *et al.* Racial/ethnic differences in circulating natriuretic peptide levels: The Diabetes Prevention Program. *PLoS One* **15**, e0229280, doi:10.1371/journal.pone.0229280 (2020).
- 16 Parcha, V. *et al.* Chronobiology of Natriuretic Peptides and Blood Pressure in Lean and Obese Individuals. *J Am Coll Cardiol* **77**, 2291-2303, doi:10.1016/j.jacc.2021.03.291 (2021).

Response to Review

Reviewer #1 (Remarks to the Author)

The authors have done an excellent job in responding to all of my comments in a positive fashion.

Many thanks.

Response: We thank the Reviewer for their positive feedback and for taking the time to review our manuscript. We appreciate the Reviewer's thoughtful comments, which helped improve the quality and clarity of our work.

Recently, a trial of withholding beta-blockers after low-risk acute myocardial infarction or withdrawal of chronic beta-blocker therapy after myocardial infarction in low risk patients have met with slightly mixed results. It would bring the discussion up to date to mention these and comment briefly.

Yndigeegn T et al. NEJM 2024

Silvain J et al. NEJM 2024

Response: We thank the Reviewer for their insightful comment. We appreciate the suggestion to discuss recent trials on the withdrawal of beta-blocker therapy post-myocardial infarction, especially in low-risk patients. Based on the Reviewer's suggestion, we have incorporated a brief summary of the findings from these recent trials in the discussion section of our manuscript.

Discussion, Page 13, Line 277

“Recent studies, including trials such as the REDUCE-AMI¹ and ABYSS² trials, examining the role of beta-blocker therapy in low-risk patients with myocardial infarction, demonstrated that discontinuing beta-blockers was not associated with a higher risk of adverse clinical outcomes in the abovementioned patient population. These findings highlight the necessity for individualizing the approach to beta-blocker therapy while taking both the cardiovascular and metabolic impacts of beta-blockers, particularly in light of their NP-augmenting effects, into account to optimize long-term health outcomes. Nonetheless, it is essential to acknowledge that the role of beta-blockers in cardiometabolic health remains complex and requires further investigation.”

Reviewer #2 (Remarks to the Author)

The authors have responded well to my initial questions. I have a few comments below. In response to the comment 2, the authors used $\text{mean} \pm \text{SD}$ as the effect size. In general, an effect size is measured as $(\text{mean difference})/(\text{SD of the difference})$. Noted that the figure shows the relationship between powers and percentage differences in means, I would suggest showing the relationship between power and effect size.

Response: We thank the Reviewer for their insightful suggestion. In response, we have updated the figure to show the relationship between power and effect size, incorporating different sample sizes for each group (**Reviewer Figure 1**). As suggested by the Reviewer, the effect size was calculated as the ratio of the mean difference to the standard deviation of the difference based on estimates from a prior study³, which resulted in an effect size of 1.47. As shown in **Reviewer Figure 1**, for the sample sizes under consideration ranging from 32 to 40, the study achieves 100% power for an effect size of 0.8 or greater.

Reviewer Figure 1: Power Curve for Detecting Mean Fold-Change in N-Terminal pro-B-type Natriuretic Peptide Levels after 6 Weeks of Metoprolol Therapy Between Black and White Individuals

Reviewer #3 (Remarks to the Author)

The authors have made much efforts to revise the comments in a point-by-point manner from my point of view. Most comments raised have been solved adequately and yet there remains 2 issues that I am not comfortable with:

1. For figure 2 (both in current and last version), I noticed that the caption (legend) mentioned about “All participants underwent standardized exercise at 70% of their maximal oxygen uptake for 20 minutes. Natriuretic peptides (NPs) were measured at baseline and immediately after exercise”.. However, the title and the text that referred to this Figure were mainly about “6 weeks of Metoprolol use Stratified by Race” rather than anything about exercise. The section about exercise, instead, should refer to Figure 1 ?

Please check these contents again carefully to ensure that correct legend was placed at the right place.

Response: We appreciate the Reviewer’s comment and apologize for this oversight. Based on the Reviewer’s suggestion, we have now corrected the Figure legend for **Figure 2**.

The updated Figure legend for **Figure 2** is as follows:

“This figure depicts the change in mid-regional pro-atrial natriuretic peptide (A), B-type natriuretic peptide (B), N-terminal proB-type natriuretic peptide (C) with 6 weeks of metoprolol stratified by race. Black and White individuals have been presented in blue and red, respectively. The values plotted depict the least square mean of natriuretic peptide and the standard error. Linear mixed models, adjusted for age, sex, body mass index, and serum insulin levels, were used to assess the racial differences in the natriuretic peptide response to metoprolol at 6 weeks.”

2. For the noticeable finding (also the key difference in this work) about “However, the NT-proBNP and MR-proANP response to exercise was found to be higher in White participants compared with Black participants”.. I expect the authors to elaborate 1-2 sentences addressing the potential implications and differences on cardiac structure/function or metabolic diseases between ethnic whites or black participants, though still speculative. This would make readers re-focus on the main study objective and clinical application of the current work.

Response: We thank the Reviewer for their suggestion. As requested by the Reviewer, we have elaborated on the potential implications of the key findings of our study and the racial differences in cardiac structure and metabolic disease.

The observed differences in NT-proBNP and MR-proANP responses to exercise between Black and White participants could have important implications for understanding racial disparities in cardiovascular and metabolic health. It is possible that the heightened NP response in White participants may reflect differences in cardiac or vascular remodeling, or in metabolic factors that influence NP release, which could predispose individuals to distinct patterns of cardiometabolic risk. In contrast, the lower NP response in Black participants might suggest a reduced capacity for adaptive cardiovascular or metabolic responses to acute physiological stress, potentially contributing to greater susceptibility to conditions like hypertension and heart failure in the long term. However, further studies are needed to explore how these differential responses might influence long-term cardiovascular and metabolic disease development and whether targeted interventions could mitigate these risks across racial groups.

References

- 1 Yndigeñ, T. *et al.* Beta-Blockers after Myocardial Infarction and Preserved Ejection Fraction. *N Engl J Med* **390**, 1372-1381, doi:10.1056/NEJMoa2401479 (2024).
- 2 Silvain, J. *et al.* Beta-Blocker Interruption or Continuation after Myocardial Infarction. *N Engl J Med* **391**, 1277-1286, doi:10.1056/NEJMoa2404204 (2024).
- 3 Davis, M. E. *et al.* Introduction of metoprolol increases plasma B-type cardiac natriuretic peptides in mild, stable heart failure. *Circulation* **113**, 977-985, doi:10.1161/CIRCULATIONAHA.105.567727 (2006).